# The Association between Use of ICS and Psychiatric Symptoms in Patients with COPD—A Nationwide Cohort Study of 49,500 Patients

**DOI:** 10.3390/biomedicines9101492

**Published:** 2021-10-18

**Authors:** Alexander Jordan, Pradeesh Sivapalan, Josefin Eklöf, Jakob B. Vestergaard, Howraman Meteran, Mohamad Isam Saeed, Tor Biering-Sørensen, Anders Løkke, Niels Seersholm, Jens Ulrik Stæhr Jensen

**Affiliations:** 1Section of Respiratory Medicine, Herlev-Gentofte Hospital, 2900 Hellerup, Denmark; pradeesh.sivapalan.02@regionh.dk (P.S.); josefin.viktoria.ekloef.01@regionh.dk (J.E.); jakob.h.vestergaard@gmail.com (J.B.V.); howraman.meteran.01@regionh.dk (H.M.); mohamad.isam.saeed.02@regionh.dk (M.I.S.); Tor.Biering-Soerensen@regionh.dk (T.B.-S.); seersholm@dadlnet.dk (N.S.); jens.ulrik.jensen@regionh.dk (J.U.S.J.); 2Department of Medicine, Hospital Lillebælt, 7100 Vejle, Denmark; aloekke@gmail.com; 3Department of Regional Health Research, University of Southern Denmark, 5000 Odense, Denmark; 4Department of Clinical Medicine, Faculty of Health and Medical Sciences, University of Copenhagen, 2200 Copenhagen, Denmark

**Keywords:** COPD, ICS, obstructive lung disease, depression, anxiety, bipolar disorder

## Abstract

Psychiatric side effects are well known from treatment with systemic corticosteroids. It is, however, unclear whether inhaled corticosteroids (ICS) have psychiatric side effects in patients with COPD. We conducted a nationwide cohort study in all Danish COPD outpatients who had respiratory medicine specialist-verified COPD, age ≥40 years, and no previous cancer. Prescription fillings of antidepressants and risk of admissions to psychiatric hospitals with either depression, anxiety or bipolar disorder were assessed by Cox proportional hazards models. We observed a dose-dependent increase in the risk of antidepressant-use with ICS cumulated dose (HR 1.05, 95% CI 1.03–1.07, *p* = 0.0472 with low ICS exposure, HR 1.10, 95% CI 1.08–1.12, *p* < 0.0001 with medium exposure, HR 1.15, 95% CI 1.11–1.15, *p* < 0.0001 with high exposure) as compared to no ICS exposure. We found a discrete increased risk of admission to psychiatric hospitals in the medium and high dose group (HR 1.00, 95% CI 0.98–1.03, *p* = 0.77 with low ICS exposure, HR 1.07, 95% CI 1.05–1.10, *p* < 0.0001 with medium exposure, HR 1.13, 95% CI 1.10–1.15, *p* < 0.0001 with high exposure). The association persisted when stratifying for prior antidepressant use. Thus, exposure to ICS was associated with a small to moderate increase in antidepressant-use and psychiatric admissions.

## 1. Introduction

Chronic obstructive pulmonary disease (COPD) is one of the leading causes of death worldwide. Inhaled corticosteroids (ICS) are used to reduce the risk of exacerbations. ICS treatment generally has fewer side-effects than treatment with oral corticosteroids (OCS). However, treatment with ICS has been associated with some of the same side effects known from systemic corticosteroid treatment, such as cataract [1,2] and pneumonia [3,4,5,6]. These side effects are thought to be either due to corticosteroids entering the bloodstream through the lungs or through the gastrointestinal tract or via a localized effect directly on the lungs. Absorption through the lungs is thought to be the primary way for ICS to reach the bloodstream, as such drugs typically have a high first-pass metabolism in the liver [7]. The extent of the side effects seems to vary between ICS type, dose, and delivery methods [7,8].

Psychiatric side effects such as depression, anxiety, bipolar disorder, and panic disorder are known from systemic corticosteroid use, with euphoria and mania being more common with short-term use and depression being more common with long-term use [9,10]. The estimated incidence of mood or anxiety disorders with systemic corticosteroid exposure varies between studies ranging from 13% to 67% [11]. One randomized trial in 50 patients reported mania in 26% in participants and depression in 10% [12]. Consistent with this, a study in patients with excessive endogenous corticosteroid production due to Cushing’s disease estimated the incidence of depression to be 56% of patients while the incidence of mania was estimated to be 31% [13]. Thus, psychiatric symptoms seem to be common with systemic corticosteroid exposure. The effects of glucocorticoids on the central nervous system are complex and poorly understood; however, the glucocorticoid receptor is widely expressed in the brain and has been shown in rats to induce fear conditioning, possibly as part of glucocorticoids’ endogenous role as a stress hormone [14]. Since ICS are known to have some systemic uptake and have an overlap with OCS in other side-effects, it could be hypothesized that ICS could to some extent be absorbed to the systemic circulation trigger the same CNS-effects as oral corticosteroids. Depression is also known to be more prevalent in COPD patients than in the general population [15], and is associated with higher rates of exacerbations [16,17], impaired quality of life [18] and increased mortality [19]. As COPD patients have an increased risk of depression and there is a theoretical basis for suspecting psychiatric side-effects to ICS treatment, it is relevant to investigate whether ICS is associated with psychiatric symptoms in patients with COPD. A small observational study found no association in patients with COPD [20]. In patients with asthma some studies have found an increase in psychiatric symptoms, while others found a decrease [21,22,23]. These differences could be explained by differences in endpoints, follow-up, and study population, thus there is a need for more studies in this area.

The aim of this study was to determine the association between ICS exposure and psychiatric symptoms effects in patients with COPD using prescription of antidepressants and psychiatric hospital admissions as a proxy for the occurrence of depression and other depressive symptoms. We hypothesized that ICS would be associated with an increase in psychiatric symptoms effects in a dose-dependent manner.

## 2. Materials and Methods

We conducted a nationwide observational cohort study in all Danish outpatients with COPD. We examined the rates of collection of antidepressant prescriptions and the rates of psychiatric hospital admissions for either depression, anxiety or bipolar disorder.

Data was obtained from the following Danish registers:(1)The Danish Register for Chronic Obstructive Pulmonary Disease (DrCOPD). A nationwide register of patients with specialist spirometry-verified COPD. The registry contains information about hospital admissions with acute COPD exacerbation and outpatient contacts. Hospitals began reporting data in 2008, and the register is monitored for consistency and completeness annually. The register contains clinical values such as forced expiratory volume in one second (FEV1), body mass index (BMI), Medical Research Council dyspnea scale (MRC) and smoking status [24].(2)The Danish National Patient Registry (DNPR). A nationwide register of all outpatient or inpatient contacts with the Danish Health Service. Each contact has a physician-coded primary diagnosis and one or more secondary diagnoses using the ICD (International Classification of Diseases), 10th revision (ICD-10). The register has received data from both the somatic and psychiatric sector since 1994 [25].(3)Danish National Database of Reimbursed Prescriptions (DNDRP). A nationwide register containing information about all collected prescriptions in community-pharmacies and hospital-based outpatient pharmacies since 2004. The register includes information on the strength, dose, product name and Anatomical Therapeutic Chemical (ATC) classification of each prescription [26].

The study population was defined as all patients registered in DrCOPD between 1 January 2010 and 31 October 2017, at least 40 years old, with no history of cancer within the last five years, except basal cell carcinoma, and at least one registered outpatient clinical visit for COPD. Patients with no outpatient contacts in DrCOPD were excluded, since we wanted to investigate outpatients. Cohort entry was defined as the day of the first outpatient clinical visit.

The accumulated ICS exposure for one year prior to cohort entry was calculated in budesonide equivalents, and the patients were grouped into four groups based on the exposure (Table A1); no ICS use, low ICS exposure, moderate ICS exposure and high ICS exposure based on the tertiles for the accumulated ICS dose. The group with no ICS exposure served as reference for the other.

The primary outcome was collection of any antidepressant prescriptions within 5 years of entering the cohort. Antidepressant doses were quantified using the number of WHO defined daily doses per year during the study period based on the ATC-codes shown in Table A2.

As a secondary outcome, we estimated the relative risk of admission to a psychiatric hospital with either bipolar disorder (ICD-10: F31), depression (ICD-10: F32–34) or anxiety (ICD-10: F40–41) within 5 years of cohort entry.

For descriptive analysis, the median and the interquartile range (IQR) for continuous variables and frequencies and proportions for categorical variables were calculated.

Inverse probability of treatment weighting (IPTW) using generalized boosted models was employed to balance covariates between the treatment groups [27]. Propensity scores were calculated based on the following confounders: GOLD 1–4 group (FEV1 percent of expected >80%, 51–80%, 31–50%, ≤30%), oral corticosteroid (OCS) use within 1 year prior to cohort entry divided into three groups (no OCS, low OCS dose and high OCS dose stratified across the median accumulated dose of 750 mg), number of exacerbations one year prior to cohort entry (0, 1, 2 or more), Body Mass Index (BMI) (continuous variable), Medical Research Council dyspnea (MRC) score (continuous variable), age (continuous variable), sex (male vs. female), active smoker (yes vs. no), Charlson comorbidity index excluding human immunodeficiency virus (HIV).

Absolute standardized mean differences (SMD) were used to assess the performance of the model. We calculated the maximum pairwise absolute SMD for each combination of the 4 treatment groups for each of our confounders.

The relative risk of prescription of any antidepressant or death by any cause was estimated using IPT-weighted Cox proportional hazards models. Cause-specific analyses were performed by right-censoring competing risks. Risk of admission to a psychiatric hospital due to depression, anxiety or bipolar disorder was analyzed in the same way. Results were presented as hazard ratios (HR) with 95% confidence intervals (CI) and cause specific HRs with 95% CI. Cumulative incidence functions for the weighted data set were calculated using the method described by Neumann et al. [28].

For sensitivity, we repeated the analysis of the risk of antidepressant use or death by any cause and risk of psychiatric admission or death by any cause while stratifying for prior antidepressant usage.

Statistical analyses were performed using SAS statistical software 9.4 (SAS Institute Inc., Cary, NC, USA) except calculation of IPTW, which was performed in R version 4.0.4 (2021-02-15), R Foundation for Statistical Computing, Vienna, Austria.

## 3. Results

We included 49,500 patients in the study. 15,420 patients had collected no ICS prescriptions one year prior to entering the cohort. A total of 34,080 patients had collected one or more prescriptions, and these patients were divided into tertiles based on their accumulated ICS dose with 11,296 patients having an accumulated dose less than 384 μg per day (low ICS exposure), 11,222 having collected between 384 μg per day and 947 μg per day (medium ICS exposure) and 11,562 having collected more than 947 μg per day (high ICS exposure) on average (Figure 1). The ICS usage did not change significantly during the 5-year follow-up period, compared to the 1-year period prior to inclusion that the groups were defined upon (Figure A1). Baseline characteristics of the groups are presented in Table 1. There were significant differences between the four groups. Patients receiving higher doses were more likely to be female, have lower BMI, higher MRC score, lower FEV1 and more frequent exacerbations. The frequencies of Charlson comorbidities were evenly distributed in all groups except for asthma, which was more common in patients receiving higher doses of ICS (Table 1). The frequency of alcohol use disorders or other substance abuse disorders was similar in all groups (Table 1).

Patients in the high ICS exposure group on average collected more anti-depressive medicine. This trend was present in all drug types except SNRI and TCA (Table A3). It is worth noting that SNRI and TCA are prescribed for neuropathic pain and psychiatric disorders [29]. Patients having collected ICS prescriptions had more admissions to psychiatric hospitals, but there was no apparent dose dependency (Table A4).

To account for differences in baseline parameters, IPTW weights for the population were calculated. Weighting reduced the absolute SMD for all confounders to less than 0.1 (Figure A2).

The risk of collection of any antidepressant or death by any cause showed a small dose- dependent association with ICS dose ((HR 1.05, 95% CI 1.03–1.07, *p* = 0.0472 with low ICS exposure, HR 1.10, 95% CI 1.08–1.12, *p* < 0.0001 with medium exposure, HR 1.15, 95% CI 1.11–1.15, *p* < 0.0001 with high exposure) as compared to no ICS exposure, Table 2 and Figure 2). The risk of admission to a psychiatric hospital or death by any cause increased in the medium and high ICS groups, but statistically insignificant in the low group when compared with the no ICS group (HR 1.00, 95% CI 0.98–1.03, *p* = 0.77 with low ICS exposure, HR 1.07, 95% CI 1.05–1.10, *p* < 0.0001 with medium exposure, HR 1.13, 95% CI 1.10–1.15, *p* < 0.0001 with high exposure). The cause-specific hazard analysis showed an increase in depressive symptoms when comparing the three ICS groups to the no ICS group, but there were no statistically significant differences among the groups (Table 2 and Figure 3).

The risk of collection of any antidepressant or death by any cause and admission with depression, anxiety or bipolar disorder or death by any cause modelled with a Cox proportional hazards model on the IPT-weighted dataset. Propensity scores for weighing were calculated based on the following confounders: GOLD 1–4 group, oral corticosteroid (OCS) use within 1 year prior to cohort entry, number of exacerbations 1 year prior to cohort entry, Body Mass Index, Medical Research Council dyspnoea score, age, gender, active smoker, Charlson comorbidity index, excluding human immunodeficiency virus. For each outcome cause specific hazards were calculated by right censoring competing risks. Abbreviations: CI: Confidence interval, HR: Hazard ratio, ICS: Inhaled corticosteroid, IPT: Inverse probability of treatment.

### Sensitivity Analyses

We repeated the main analyses while stratifying for prior antidepressant use within 5 years of entering the cohorts. We saw slightly decreased hazard ratios for both outcomes, but the signal persisted. Risk of collection of antidepressants or death by any cause: HR 1.03, 95% CI 1.01–1.05, *p* = 0.0008 with low ICS use, HR 1.06, 95% CI 1.04–1.08, *p* < 0.0001 with medium use, HR 1.14, 95% CI 1.04–1.17, *p* < 0.0001 with high use. Risk of psychiatric admission or death by any cause: HR 1.00, 95% CI 0.98–1.02, *p* = 0.99 with low ICS use, HR 1.06, 95% CI 1.04–1.09, *p* < 0.0001 with medium use, HR 1.12, 95% CI 1.10–1.15, *p* < 0.0001 with high use.

As TCA and SNRI are also prescribed to treat neuropathic pain, we repeated the main analysis using any antidepressant other than TCA and SNRI or all-cause mortality as the endpoint. This slightly increased the signal: HR 1.05, 95% CI 1.03–1.07, *p* < 0.0001 with low ICS use, HR 1.12, 95% CI 1.10–1.14, *p* < 0.0001 with medium use, HR 1.19, 95% CI 1.16–1.21, *p* < 0.0001 with high use.

A large majority of patients receiving ICS received either budesonide (52.8%) or fluticasone (45.7%). We performed a sensitivity analysis comparing these two drugs in relation to the main outcomes, but did not find any significant differences.

## 4. Discussion

In these nationwide data from COPD outpatients over nearly eight years and with complete follow-up and control for known and possible confounders, we observed a small dose-dependent association between ICS usage in COPD patients and antidepressant use. This signal persisted after stratifying for prior antidepressant usage. There was a similar increase in the risk of hospital admission for depression, anxiety or manic disorder, but there was no clear dose dependence for this outcome. Worth noting is that there were few hospital admission events in our population, and the lack of dose-dependence could be because our study was insufficiently powered to distinguish the three groups receiving ICS.

We identified a single study that examined the association between psychiatric symptoms and ICS exposure in COPD patients, which found no association. This study was smaller and had a shorter 12-week follow-up period, which may explain why they did not detect a signal [20]. It may also be the case that our small signal is due to bias by indication. In patients with asthma, a small study from 2003 found an association between depressive symptoms and increasing ICS dose, but it is difficult to say if this was due to ICS treatment or only reflected the severity of the asthma, as the study was small and did not control for all important confounders [21]. Another small study in asthma patients from 2006 reported behavioral alterations such as increased aggression or anxiety during ICS treatment [22]. A randomized study found that long-term treatment with a low dose of ICS (200 μg budesonide per day) lowered depressive symptoms in children with Asthma compared to placebo [23]. This dosage is smaller than the cutoff for the lowest ICS group in our study, so the differences could be due to the low dose [23]. Assuming that the observed increase in depressive symptoms was entirely caused by the ICS treatment, the low magnitude of the increase indicates a low clinical significance of our finding. The benefit from reducing exacerbations and symptoms would outweigh the slight increase in relative risk of psychiatric events.

### Strengths and Limitations

Our study followed a large well-defined nationwide cohort of COPD patients with complete data follow-up for collection of prescriptions and hospital admissions [24,25,26]. We also had access to a complete set of prescription data, as well as precise data on possible confounders of a signal, like FEV1, BMI and smoking status.

There are, however, noteworthy limitations to our study. First, since this is an observational study, it is not possible to conclude any causal relationship between ICS exposure and depressive symptoms. Second, there were significant baseline differences in our population and patients receiving more ICS in general had more severe diseases. While we attempted to adjust for this, residual confounding cannot be ruled completely out and may play a significant role in the small signal we do observe. We tried to account for this in our sensitivity analysis by stratifying for prior antidepressant use, which did not change the signal. In addition, there was a small dose-dependent increase in IPTW all-cause mortality, which could indicate some residual confounding. Third, information about medicine usage was obtained based on collected prescriptions. This gives no information about patient compliance, and it is a possibility that some patients did not fully or correctly use the prescriptions they collected. This would affect both the primary endpoint of antidepressant use and our ICS grouping. Fourth, we do not have any information on the indication for the antidepressant prescriptions. Thus, antidepressant prescriptions could indicate a range of conditions such as depression, anxiety, pain and insomnia. Especially TCA and SNRI are commonly prescribed for neuropathic pain [29,30]. For this reason, we conducted a sensitivity analysis excluding TCA and SNRI, which did not alter the signal, but this does not account for other indications in other drug groups. Fifth, the frequency of psychiatric admissions was very low, and while we saw an increase in risk with ICS usage relative to no ICS usage, we did not see any significant differences between ICS groups. Our study may have been insufficiently powered to compare hospital admissions between patients receiving ICS.

In conclusion, ICS use was associated with a small dose dependent increase in risk of antidepressant use and a small increase in risk admission to psychiatric hospital. The magnitude of this possible increase is thus not high and regarding depressive symptoms, ICS drugs do seem safe in most patients.

## Figures and Tables

**Figure 1 biomedicines-09-01492-f001:**
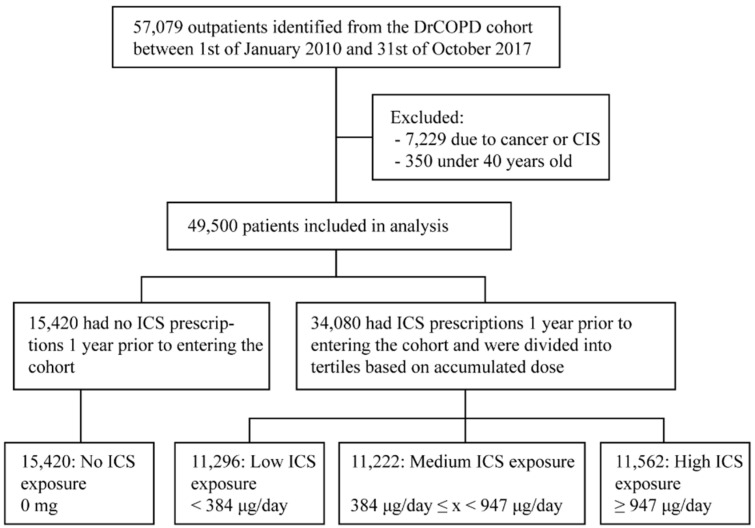
Flow diagram for the study population. Patients with at least 1 outpatient clinical visit between 1st of January 2010 and 31st of October 2017 were included from the Danish register for COPD (DrCOPD). Patients younger than 40 years old or with cancer other than basal cell carcinoma of the skin were excluded. Patients were divided into four groups on the basis of the accumulated ICS dose one year prior to entering the cohort. Abbreviations: CIS: carcinoma in situ. ICS: Inhaled corticosteroid.

**Figure 2 biomedicines-09-01492-f002:**
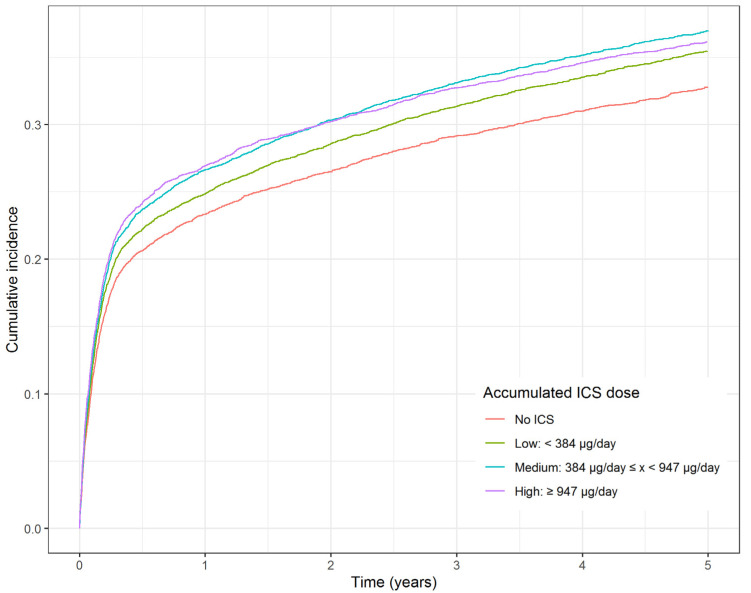
Cumulative incidence function for collection of antidepressants within 5 years using inverse probability of treatment weighing. Abbreviations: ICS: Inhaled corticosteroid.

**Figure 3 biomedicines-09-01492-f003:**
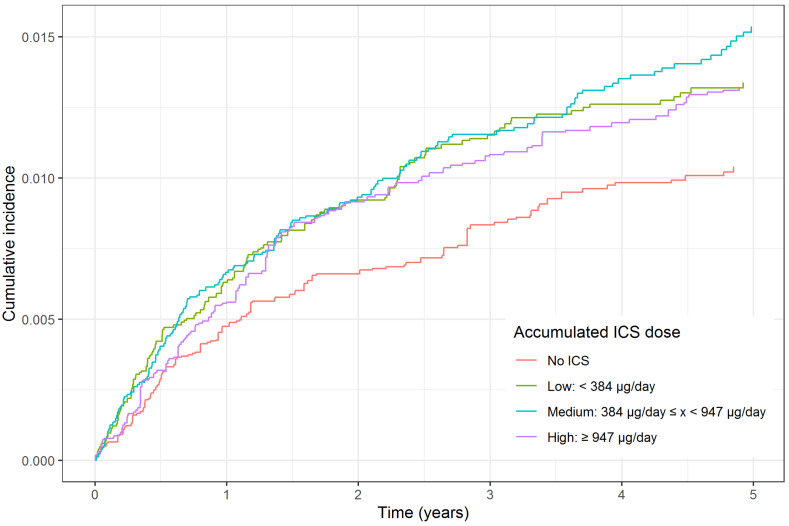
Cumulative incidence function for hospital admission with either depression, anxiety or bipolar disorder using inverse probability of treatment weighing. Abbreviations: ICS: Inhaled corticosteroid.

**Table 1 biomedicines-09-01492-t001:** Baseline characteristics of the COPD outpatients between 1 January 2010 and 31 February 2017.

	No ICS Exposure N = 15,420	Low ICS Exposure N = 11,296 <384 μg/day	Medium ICS Exposure N = 11,222 384 μg/day ≤ x <947 μg/day	High ICS Exposure N = 11,562 ≥947 μg/day
Age, median (IQR)	69 (60.75–76)	70 (61–77)	71 (64–78)	71 (64–78)
Male, n (%)	8102 (52.5)	5525 (48.9)	5102 (45.5)	4728 (40.9)
Smoking				
Current smoker, n (%)	5793 (37.6)	3887 (34.4)	3369 (30.0)	3435 (29.7)
Former smoker, n (%)	7257 (47.1)	6254 (55.4)	6994 (62.3)	7347 (63.5)
Never smoker, n (%)	306 (2.0)	188 (1.7)	146 (1.3)	104 (0.9)
Smoking status unknown, n (%)	2064 (13.4)	967 (8.6)	713 (6.4)	676 (5.8)
Diagnosed with alcohol use disorder, n (%)	920 (6.0)	619 (5.5)	512 (4.6)	513 (4.4)
Diagnosed with substance abuse disorder other than alcohol, n (%)	869 (5.6)	601 (5.3)	487 (4.3)	501 (4.3)
BMI, median (IQR), kg·m^−2^	25 (22–29)	25 (22–29.3)	25 (21–29)	24 (21–28)
<18.5, n (%)	1085 (7.0)	800 (7.1)	1013 (9.0)	1404 (12.1)
18.5–24.9, n (%)	4952 (32.1)	3770 (33.4)	4062 (36.2)	4475 (38.7)
25–29.9, n (%)	4174 (27.1)	3182 (28.2)	3130 (27.9)	2941 (25.4)
30–34.9, n (%)	2071 (13.4)	1639 (14.5)	1512 (13.5)	1345 (11.6)
>=35, n (%)	1153 (7.5)	886 (7.8)	779 (6.9)	730 (6.3)
MRC, median (IQR)	2 (2–3)	3 (2–4)	3 (2–4)	3 (3–4)
1, n (%)	7130 (46.2)	6193 (54.8)	5761 (51.3)	5533 (47.9)
2, n (%)	9796 (63.5)	8275 (73.3)	7635 (68.0)	7041 (60.9)
3, n (%)	8709 (56.5)	8248 (73.0)	8539 (76.1)	8382 (72.5)
4, n (%)	6921 (44.9)	6855 (60.7)	7391 (65.9)	8022 (69.4)
5, n (%)	6005 (38.9)	6026 (53.3)	6563 (58.5)	7299 (63.1)
FEV_1_ % predicted, median (IQR)	58 (44–70)	52 (40–65)	45 (33–58)	39 (29–52)
>=80, n (%)	1642 (10.6)	777 (6.9)	431 (3.8)	261 (2.3)
50–79, n (%)	7218 (46.8)	4890 (43.3)	3795 (33.8)	2904 (25.1)
30–49, n (%)	3636 (23.6)	3597 (31.8)	4362 (38.9)	4781 (41.4)
<30, n (%)	872 (5.7)	1006 (8.9)	1860 (16.6)	2867 (24.8)
Acute exacerbations requiring hospital admission in the past year, n (%)				
0	13215 (85.7)	8219 (72.8)	7711 (68.7)	7394 (64.0)
1	1096 (7.1)	1412 (12.5)	1410 (12.6)	1678 (14.5)
2 or more	1109 (7.2)	1665 (14.7)	2101 (18.7)	2490 (21.5)
Charlson comorbidity index excluding HIV, median (IQR)	4 (3–5)	4 (3–5)	4 (3–5)	4 (3–5)
Myocardial infarction, n (%)	1145 (7.4)	870 (7.7)	848 (7.6)	813 (7.0)
Heart failure, n (%)	2396 (15.5)	1792 (15.9)	1675 (14.9)	1735 (15.0)
Peripheral vasc. disease, n (%)	1943 (12.6)	1343 (11.9)	1274 (11.4)	1219 (10.5)
Cerebrovasc. disease, n (%)	1860 (12.1)	1367 (12.1)	1248 (11.1)	1180 (10.2)
Dementia, n (%)	265 (1.7)	162 (1.4)	177 (1.6)	192 (1.7)
Rheumatic disease, n (%)	792 (5.1)	509 (4.5)	470 (4.2)	411 (3.6)
Peptic ulcers, n (%)	779 (5.1)	601 (5.3)	554 (4.9)	645 (5.6)
Mild liver disease, n (%)	462 (3.0)	296 (2.6)	217 (1.9)	224 (1.9)
Moderate or severe liver disease, n (%)	109 (0.7)	59 (0.5)	49 (0.4)	34 (0.3)
Diabetes mellitus without chronic complication, n (%)	1874 (12.2)	1347 (11.9)	1166 (10.4)	1200 (10.4)
Diabetes mellitus with chronic complication, n (%)	645 (4.2)	452 (4.0)	369 (3.3)	327 (2.8)
Kidney disease, n (%)	729 (4.7)	469 (4.2)	374 (3.3)	338 (2.9)
Asthma, n (%)	1013 (6.6)	1728 (15.3)	2106 (18.8)	2481 (21.5)

Abbreviations: BMI: body mass index; MRC: Medical Research Council dyspnoea scale; FEV_1_: forced expiratory volume in 1 s.

**Table 2 biomedicines-09-01492-t002:** Outcomes within 5 years of cohort entry.

	No ICS Exposure N = 15,420	Low ICS Exposure N = 11,296 <384 μg/day	*p*-Value	Medium ICS Exposure N = 11,222 384 μg/day ≤ x <947 μg/day	*p*-Value	High ICS Exposure N = 11,562 ≥947 μg/day	*p*-Value
Collection of any antidepressant within 5 years or all-cause mortality, HR (95% CI)	Reference	1.05 (1.03–1.07)	0.0472	1.10 (1.08–1.12)	<0.0001	1.15 (1.11–1.15)	<0.0001
Collection of any antidepressant, HR (95% CI) ^a^	Reference	1.10 (1.07–1.13)	<0.0001	1.16 (1.13–1.19)	<0.0001	1.17 (1.15–1.20)	<0.0001
All-cause mortality, HR (95% CI) ^a^	Reference	1.00 (0.97–1.02)	0.77	1.06 (1.04–1.09)	<0.0001	1.12 (1.10–1.15)	<0.0001
Admission with diagnosis depression, anxiety or bipolar disorder or all-cause mortality, HR (95% CI)	Reference	1.00 (0.98–1.03)	0.77	1.07 (1.05–1.10)	<0.0001	1.13 (1.10–1.15)	<0.0001
Admission with diagnosis depression, anxiety or bipolar disorder, HR (95% CI)^a^	Reference	1.21 (1.07–1.38)	0.0035	1.32 (1.16–1.49)	<0.0001	1.22 (1.07–1.39)	0.0027
All-cause mortality, HR (95% CI)^a^	Reference	1.00 (0.97–1.02)	0.77	1.06 (1.04–1.09)	<0.0001	1.12 (1.10–1.15)	<0.0001

^a^ Cause specific hazards were calculated by right-censoring competing risks.

## Data Availability

We believe that knowledge sharing increases the quality and quantity of scientific results. The sharing of relevant data will be discussed within the study group upon reasonable request.

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
