# Peer review of "The Association between Use of ICS and Psychiatric Symptoms in Patients with COPD—A Nationwide Cohort Study of 49,500 Patients"

_biomedicines, 2021, doi:10.3390/biomedicines9101492_

Round 1

Reviewer 1 Report

I read with interest the study of Jordan et al. on the association of ICS and psychiatric symptoms in COPD. The data are clear and well-presented, I only have some comments. 

  1. The ICS dose was calculated from data of 1 year, but psychiatric outcomes are measured for 5 years. Was the ICS dose constant for 5 years?
  2. Did the authors find an effect of the different ICS classes on the main outcomes?
  3. It is not clear how ICS could induce or modify the course of psychiatric symptoms. Please discuss it in more detail. 
  4. Did the patients have other diseases such as alcohol or either substance abuse which could modify the psychiatric symptoms?  
  5. Please correct the typo in "myocardial infarction" in the table. 

Author Response

C1: The ICS dose was calculated from data of 1 year, but psychiatric outcomes are measured for 5 years. Was the ICS dose constant for 5 years?

R_C1: Thanks for pointing this out as it is important in the interpretation of our study. The ICS dose is quantified one year prior to baseline, while the outcomes are observed in the 5 years after baseline. We have now calculated the median dose for each of the four treatment groups and added an extra figure that shows that the ICS exposure did not change significantly throughout the observation period.

C2: Did the authors find an effect of the different ICS classes on the main outcomes?

R_C2: Thank you for asking this interesting question. As more than 98% of the patients in the study received either budesonide or fluticasone we specifically compared these two classes of ICS. We did not find any significant differences.

C3: It is not clear how ICS could induce or modify the course of psychiatric symptoms. Please discuss it in more detail.

R_C1: Thanks for this suggestion. We have elaborated on this in our introduction by discussing the frequency of psychiatric side-effects with oral corticosteroids in addition to citing studies showing that ICS can enter the bloodstream and exhibit systemic effects. We believe that if ICS were to have psychiatric effects it would be through absorption to the bloodstream and through a mechanism like that of oral corticosteroids. The molecular effects of on the central nervous system are not well known, but the psychiatric effects could be a consequence of the endogenous role of glucocorticoids as a stress hormone.

We have included the following text in the introduction of the manuscript:

The estimated incidence of mood or anxiety disorders with systemic corticosteroid exposure varies between studies ranging from 13% to 67% [11]. One randomized trial in 50 patients reported mania in 26% in participants and depression in 10% [12]. Consistent with this, a study in patients with excessive endogenous corticosteroid production due to Cushing’s disease estimated the incidence of depression to be 56% of patients [13] while the incidence of mania was estimated to be 31% [13]. Thus, psychiatrics symptoms seem to be common with systemic corticosteroid exposure. The effects of glucocorticoids on the central nervous system are complex and poorly understood, however, the glucocorticoid receptor is widely expressed in the brain and has been shown in rats to induce fear conditioning, possibly as part of glucocorticoids endogenous role as a stress hormone [14]. Since ICS are known to have some systemic uptake and have an overlap with OCS in other side-effects, it could be hypothesized that ICS could to some extent be absorbed to the systemic circulation trigger the same CNS-effects as oral corticosteroids

C4: Did the patients have other diseases such as alcohol or either substance abuse which could modify the psychiatric symptoms?  

R_C4: Thanks for asking this question. We have included figures from the Danish national patient registry about the rates of alcohol abuse disorders and other substance abuse disorders. These do not differ significantly between the 4 groups of the study.

The following information has been included in the baseline table (table 1)

Diagnosed with alcohol use disorder, n (%)

920 (6.0)

619 (5.5)

512 (4.6)

513 (4.4)

Diagnosed with substance abuse disorder other than alcohol, n (%)

869 (5.6)

601 (5.3)

487 (4.3)

501 (4.3)

C5: Please correct the typo in "myocardial infarction" in the table.

R_C5: Thanks for noticing this mistake. It has now been corrected.

Reviewer 2 Report

There have been related studies on inhaled corticosteroids (ICS) and mental side effects in COPD patients. The article pointed out that there is no significant difference between emotional problems and inhalation therapy in COPD patients (Hyun, Lee et al. 2016). However, this study specifically targeted the use of ICS in Danmark COPD outpatients to be associated with antidepressants and psychosis.

Some questions are as follows

  1. Please reconfirm the number of people in Figure 1. The number of outpatients deducting cancer patients and patients under the age of 40 from the remaining analysis is not correct.

  1. This study has a clear definition of the inhaled corticosteroids (ICS) dose to distinguish the tested population. It is recommended that a clear definition of the population in the supplementary anxiety and the use of antidepressant drugs

  1. It is recommended to add Hospital Anxiety and Depression Scale (HADS)

  1. Line 197 Table 1. Baseline characteristics of the COPD outpatients data, because COPD and smoke are related, it is recommended to add secondhand smoke information

References:

Hyun MK, Lee NR, Jang EJ, Yim JJ, Lee CH. Effect of inhaled drugs on anxiety and depression in patients with chronic obstructive pulmonary disease: a prospective observational study. Int J Chron Obstruct Pulmon Dis. 2016 Apr 11;11:747-54. doi: 10.2147/COPD.S96969. PMID: 27114705; PMCID: PMC4833365.

Author Response

C1: Please reconfirm the number of people in Figure 1. The number of outpatients deducting cancer patients and patients under the age of 40 from the remaining analysis is not correct.

R_C1: Thanks for noticing this mistake. The discrepancy has now been corrected.

C2: This study has a clear definition of the inhaled corticosteroids (ICS) dose to distinguish the tested population. It is recommended that a clear definition of the population in the supplementary anxiety and the use of antidepressant drugs

R_C2: We agree that this improves the clarity of the tables, so we have updated the headers to include this information. Thanks for this suggestion.

C3: It is recommended to add Hospital Anxiety and Depression Scale (HADS)

R_C3: Thank you for this suggestion. Unfortunately, this information is not accessible neither in the general registries on the Danish population nor in our national COPD registry, so we do not have access to this information and therefore cannot include it in our analysis. For prospective studies, we shall remember to include this important scale.

C4: Line 197 Table 1. Baseline characteristics of the COPD outpatients data, because COPD and smoke are related, it is recommended to add secondhand smoke information

R_C4: Almost the entire population (98%) have been, or are, active smokers. We have included information on smoking status into the baseline table. Unfortunately, we do not have information about secondhand smoke.

We have included the following information in the baseline table (table 1).

Smoking

Current smoker, n (%)

5793 (37.6)

3887 (34.4)

3369 (30.0)

3435 (29.7)

Former smoker, n (%)

7257 (47.1)

6254 (55.4)

6994 (62.3)

7347 (63.5)

Never smoker, n (%)

306 (2.0)

188 (1.7)

146 (1.3)

104 (0.9)

Smoking status unknown, n (%)

2064 (13.4)

967 (8.6)

713 (6.4)

676 (5.8)

Diagnosed with alcohol use disorder, n (%)

920 (6.0)

619 (5.5)

512 (4.6)

513 (4.4)

Best regards

Alexander Jordan

alexander.svorre.jordan@regionh.dk

Round 2

Reviewer 2 Report

The author has replied to all my questions, no further comments